# Assessing the Invasion Risk of *Humulus scandens* Using Ensemble Species Distribution Modeling and Habitat Connectivity Analysis

**DOI:** 10.3390/plants11070857

**Published:** 2022-03-23

**Authors:** Mariana Mihaela Urziceanu, Alina Georgiana Cîșlariu, Eugenia Nagodă, Alma Lioara Nicolin, Dragoș Ștefan Măntoiu, Paulina Anastasiu

**Affiliations:** 1Department of Botany and Microbiology, Faculty of Biology, University of Bucharest, 1-3 Intr. Portocalelor, 060101 Bucharest, Romania; mariana-mihaela.urziceanu@bio.unibuc.ro (M.M.U.); paulina.anastasiu@bio.unibuc.ro (P.A.); 2Garden “D. Brandza”, University of Bucharest, 32 Sos. Cotroceni, 060114 Bucharest, Romania; eugenia.nagoda@bio.unibuc.ro; 3Biology—Plant Sciences, Banat University of Agricultural Sciences and Veterinary Medicine “King Michael I of Romania”, 300645 Timisoara, Romania; alma_nicolin@usab-tm.ro; 4“Emil Racoviță” Institute of Speleology—Cluj Department, 5 Clinicilor st., 400006 Cluj-Napoca, Romania; stephen.mantoiu@gmail.com

**Keywords:** invasiveness, alien plants, distribution, modeling, EU concern, Romania

## Abstract

Given the rapid spread of invasive alien plant species in Europe and limited information regarding their distribution and dispersion patterns, we analyzed the invasive risk of *Humulus scandens*, a species with an increased invasive potential. We collected occurrence records from Romania within an EU funded project and literature data, in order to perform an ensemble distribution model. Environmental variables varied from downscaled topoclimatic continuous entries to categorical ones, such as soil class, texture, or land use. Results showed potential core areas of the species within the study region. By inverting the probability output of the models, we have created a resistance surface which helped us model its dispersion patterns. Further, we assessed the probability of invasion for each resulted corridor using the species dispersion ecology and created an invasion risk map. *H. scandens* is highly influenced by milder climates and areas with constant flooding events, thus we found that the Tisa basin and its tributaries can be under a high invasion risk, spreading through the entire catchment, in Central, Western, and Northern Romania, towards the Eastern Carpathians. The Danube acted as a dispersion corridor for major river systems in southern Romania, but the dispersion capability of the species dropped in steppe areas with higher aridity and limited water course network. This approach is useful for creating adequate action plans in relation to invasive alien plant species, and should urgently be regarded, as results show a potentially large distribution of *H. scandens* across entire water catchment areas, with devastating effects on natural ecosystems.

## 1. Introduction

Assessing and controlling the invasive risk of alien plant species have become essential management procedures for biodiversity conservation. The invasive alien plant species (IAPs) can disrupt entire ecosystems, with cascading long-term effects such as massive biodiversity loss and soil degradation, sometimes causing irreversible damage to the environment and its services [1,2,3,4,5].

Managing IAPs is a priority for the European Union, as mentioned in the Biodiversity Strategy for 2030 [6], according to which, the invasive alien species (IAS) are subject to the EU Regulation No 1143/2014, which aims to prevent and limit their negative effects on the environment. One of the most important measures within this regulation refers to the development of a surveillance system for all member states, which should include up-to-date distribution data on IAS, early detection mechanisms, spread pattern recognition systems and invasion risk assessments. Distribution data for IAPs are insufficient at the scale and spatial resolution required to identify areas at high risk of invasion [7]. Therefore, periodic updates are being made to the list of EU-IAS as more data is collected [8], especially for the species which can affect vulnerable natural habitats.

Wetlands and riverbank habitats are extremely vulnerable to invasion risk because they mostly have a linear or concentrated distribution that makes them more prone to edge effects [9]. Furthermore, running water represents an optimal dispersion environment for IAPs, offering optimal pathways and endangering connected suitable natural habitat [10], sometimes regardless of distance or even water flow direction [11,12].

Given the limited data regarding the distribution and invasion risk patterns of recently included species within the list of the EU-IAS, the current management recommendations of the regulation for member states, and the sensitivity of wetland areas with regard to IAPs, we considered assessing the invasion risk and current distribution of a species with a high dispersion ability: *Humulus scandens*. Our decision was also supported by the findings of a conservation project implemented within the EU Regulation framework (Adequate management of invasive species, POIM2014 + 120008) in Romania, in which systematic field surveys revealed a much wider spread of the species than earlier expected, further increasing the need for an invasion risk assessment.

*H. scandens* is a climbing, fast-growing herbaceous plant species, with an eastern Palearctic origin [13], which was introduced and became invasive in Europe [14] and North America [15]. Its dispersion is ensured by seeds which are easily transported by water [13]. This plant can act also as a perennial, producing adventitious roots in response to stress conditions, such as flooding [16]. However, reproduction occurs through seeds that can survive in the soil for about three years [17], transport of gravel with seeds being considered a possible source of infection of other habitats [18]. The seeds germinate in early spring, peaking to the formation of about 35 seedlings per m^2^ in March [16,19], subsequently having a significant growth and a strong competitiveness [20]. Once installed, it forms dense clusters, obstructing light for native plant species and changing the structure of wetlands and riverbeds [14,21]. It can also colonize open spaces and transitional habitats with some edge effect elements, such as road and railway sides, or even grasslands and forest edges [22]. The species is also found as a decorative plant in some major retail markets across Europe, which furthers its dispersion potential. In Europe, *H. scandens* has become invasive in southern France [16], Hungary [13], Italy [23], and Northern Croatia [24]. It is subspontaneous in the Pannonian plain, Serbia [25], and Bulgaria [26], and has been rarely recorded in Austria [27], Belgium [28], Czech Republic [29], Germany [13], Denmark [30], Poland [30], Slovenia [13], Slovakia [31], Sweden [32], Switzerland [13], Ukraine [33], and until now, in Romania [34,35].

In Romania, the species was first observed in the wild in 1942 [36], in disturbed areas from the southern region, with no information regarding its origins [35,36,37]. Several other occurrences from the west and south-west regions were recorded along the Danube and some of its tributaries [38,39,40]. Later observations were collected from north-western Romania, along water channels and shrubbery forest edges [41,42]. Based on these references, *H. scandens* was classified as a casual species in Romania [34,35], yet our recent findings show a much larger distribution than previously reported.

For a better understanding of the drivers that lead to the biological invasion, SDMs (species distribution models) have been increasingly used to predict IAS distribution [43,44,45,46]. Furthermore, SDMs are widely used for establishing reliable biodiversity conservation and management strategies [47,48,49,50]. This method is facilitating the early detection of new invasions and maximizes the success of risk assessment protocols [51,52,53,54,55]. In addition, dispersal also shapes species distributions, as they will colonize suitable, available geographic space [56,57,58,59]. Combining SDMs with dispersal modeling can improve range dynamics, which are influenced by climate and dispersal [59].

Ecological reconstruction projects require considerable resource investments and may have limited effects on the restoration of degraded natural habitats affected by IAPs, losing valuable ecosystem services in the process. Therefore, prevention and timely action plans can be the most cost-effective solutions [5,60,61,62], with a potential starting point in SDM approaches [63].

Since prevention approaches are considered the most effective strategies in limiting IAPs dispersion [64,65,66], our study aims to update the species distribution from Romania, identify the other suitable habitats of *H. scandens* using SDMs, identify its potential dispersion pathways and create an invasion risk map which can be used as a tool for management practices. Given the high dispersion ability of *H. scandens* and the systematic occurrence data we collected within the Project “Adequate management of invasive species, POIM2014 + 120008“, we assessed the invasion risk of this species by using SDMs and habitat connectivity analysis.

## 2. Results

Data recorded in the field showed a much wider distribution of the species in Romania than previously known, with considerable other potentially suitable habitats across the study area (Figure 1).

The ensemble SDM registered optimal values for all the evaluation metrics, showing a good predictive performance (Table 1). Only half of the modeling algorithms registered AUC values greater than 0.75 (Table 1). The random forest model (RF) registered the highest values of Sensitivity, Specificity, AUC, TSS, as well as the lowest values of the omission rate (Table 1).

The variables with the highest contribution in modeling the potential suitability of occurrence of *H. scandens* were the soil texture, land use type, distance to roads and railways, Bio6, and distance to water courses (Table 2).

The ensemble SDM shows that the species meets its environmental requirements especially in the southern and western regions of Romania, spreading mainly along the main water courses (Figure 1). A lower habitat suitability was observed in some areas from north—north-western and central Romania (Figure 1). These results are in accordance with our field research, *H. scandens* being identified with a high frequency in the southern and south-western regions of Romania.

The model also displayed some topoclimatic suitable sites within the eastern region of Romania where the species has not yet been identified from the field surveys, or literature review.

The habitat connectivity analysis showed the potential dispersion pathways of *H. scandens* across Romania and was further processed to assess its invasion risk (Figure 2).

The habitat suitability analysis resulted in 960 cores with surfaces above the 1 km^2^ threshold, from which 20 were confirmed according to personal data and literature review.

The connectivity analysis generated 1382 dispersion corridors, each containing a single LCP (least cost path) line. The invasion risk classification resulted in 1004 high invasion risk LCPs (total distance: 16,330.17 km, average distance: 16 km per LCP, maximum distance: 308 km, minimum distance: 2.5 km), 339 medium invasion risk LCPs (total distance: 17,497.23 km, average distance: 51 km per LCP, maximum distance: 439 km, and minimum distance: 5 km), and 39 low invasion risk LCPs (total distance: 5353.92 km, average distance: 137 km, maximum distance: 295 km, minimum distance: 41 km). Only 36 LCPs were directly connected to the 20 confirmed core areas (32—high risk, 2—medium risk, and 2—low risk).

The most abundant corridors resulting from the connectivity analysis were in the western area. The rivers in this region are tributary to the Tisa River which drains the western, central, and northern parts of Romania. The highest abundance of high-risk LCPs was also registered in the western part of Romania. The central region was connected only through several LCPs, but also contained validated core areas. The Danube River acted as a continuous corridor with extensions on the major river basins in the southern region. The eastern and north-eastern regions were poorly connected despite the existence of potentially suitable habitats.

## 3. Discussion

*H. scandens* has a high invasion risk in Europe, spreading through major river systems, as well as through ongoing cultivation and retail distribution [12]. Given our recent observations regarding the expansion of the species range in Romania, associated with a potentially increased risk of invasiveness, we assessed its potential distribution to detect other possible unidentified suitable habitats based on its ecological requirements and afterwards performed an invasive risk analysis.

Since *H. scandens* occurs mainly in wet habitats, on riversides and floodplains, it has been considered a plant of clayey-loam soil [13]. In its native range, it is also predominantly established on loamy-sandy ground [13]. We therefore considered the soil texture as a variable that may influence the distribution of *H. scandens* and, as expected, it registered an important contribution to the ensemble model. In Romania, as well as in other European countries and in its native range, *H. scandens* was found mainly on loamy, loamy-clay, or loamy sand-clay soils, which further highlights its preferences for the moist areas. We also considered the soil class as a variable potentially influencing the species distribution. Even though it did not register a high contribution to the ensemble model (Table 2), it is worth mentioning that the anthrosol, which is the dominant soil class occupied by *H. scandens* in Romania, along with the preferred soil texture, may confirm that the species prefers wet, alluvial habitats with loamy sand-clay soils, especially if they are disturbed. The habitats with *H. scandens* in Romania are similar to previous records from other European countries where it is invasive, highlighting its preferences for the alluvial, disturbed areas [13,24,26]. The alluvial habitats are preferred not only by *H. scandens* but also by other invasive or potentially invasive lianas in Europe. Invasive climbing species may be even stronger competitors with a more successful invasion rate, thus their impact on habitats and biodiversity is a matter of serious ecological concern [67]. *Echinocystis lobata* is one of the most abundant invasive climbing species from Europe, spreading in natural riparian forests, thickets, and tall herbs [68]. *Sicyos angulatus*, another invasive liana in Europe, also prefers riparian habitats where the soils are wet and fertile [69]. During our fieldwork, we observed these two lianas in some of the identified phytocoenoses with *H. scandens*, which further exacerbates the degradation of the riparian habitats. This highlights once again the necessity to establish and implement effective management measures to avoid their farther deterioration. *Celastrus orbiculatus*, a potentially invasive liana in Europe, has also the ability to establish in alluvial and riparian habitats and spread along watercourses [67]. Therefore, the alluvial habitats can be further considered very useful routes for the invasive plants’ species dispersion.

The land use type is the second most important predictor variable which models the distribution of *H. scandens* in Romania. The species mainly inhabits disturbed areas near roadsides, settlements, forests, shrub lands edges, and dump sites (Appendix A). The species was recorded from the same land use types in other European countries [12,13]. We consider this as another human-impact related variable that features its preferences for the disturbed areas.

Since most of our recordings, as well as those from the literature, were registered along roadsides and riverbanks, we considered the distance to roads and infrastructure and the distance from watercourses as variables that may influence the species distribution across Romania. As expected, the first variable registered a high contribution to the ensemble model.

We assumed that the distance from watercourses would have a higher influence on the species distribution than our ensemble SDM registered, since the water availability is a limiting parameter with a significant impact on the distribution and expansion of species [20]. Similar to our results, the distance to the river was not a decisive factor corresponding to *H. scandens* success in France (Gordon River) [16]. Nevertheless, it is worth mentioning that a narrow ecological niche and a reduced competitiveness during prolonged water stress of *H. scandens* may suggest that environments near rivers are the most vulnerable to the invasion of this species [20]. From the initially selected bioclimatic variables, only the mean minimum temperature of the coldest month (Bio6) registered a high contribution to the distribution of *H. scandens* in Romania. We considered this result as relevant because low temperatures recorded during the vegetation season are a limiting factor of *H. scandens* spread in Europe, since the seeds require cold stratification to germinate [12,13]. We also considered including several other climatic variables for the ensemble modeling. These reflected temperature fluctuations, especially the temperature during the driest months [62,70,71] and precipitation variability [72,73], because it has been observed that increasing precipitation seasonality restricts the spread of the invasive plant species [62]. In the end, the mentioned variables were either excluded from the analysis because of the collinearity with the other predictor variables or registered a low contribution to the ensemble SDM.

Given the structured bioclimatic influences which are found in Romania, the species is most likely able to colonize the western areas, because they register milder climates with more abundant precipitations compared to the eastern steppe regions [74]. The steppe bioregion (east and south-eastern Romania) is also most likely unsuitable for its rapid expansion because of a less developed hydrographic network.

The connectivity analysis confirmed that the western region has the greatest invasion risk, using the Tisa and its major tributaries as dispersion pathways. The topoclimatic conditions, as well as its high dispersion capability facilitated by seasonal floods contribute to the high invasion risk of the species within this region. The inundation areas are most exposed to invasion, where even a single individual can spread over long distances [13]. The Danube also facilitates dispersion in the southern area, on both sides of the river, especially on its most important tributaries. Further monitoring within the mid and lower Danube basin is required in order to control the species rapid expansion.

The hydrographic network of the eastern region follows a predominantly N-S direction, draining a large plateau. The entire NE region is connected to the Danube by two main tributaries (Prut and Siret rivers), making it harder for the species to disperse, in contrast to the southern region, where each major tributary has a direct connection to the Danube. Within the NE area, the model found less suitable areas for the species, with a lower invasion risk, most likely because precipitation is less abundant compared to the western region. The SE part of Romania (Dobrogea) yielded no significant results because of its high aridity index and limited hydrographic network.

Lower abundance of dispersal corridors was observed in central Romania, concentrated on major river systems. The species was present at higher altitudes within these basins (Someș, Mureș), suggesting that it can actually have a much wider distribution along the identified corridors. Therefore, further field research is necessary in order to confirm its presence in the area. Our analysis showed that corridors can exist between major water catchments but were classified as low invasion risk because they cross multiple basins, over long distances. However, one such corridor has been identified between the central and eastern parts of Romania (in the Eastern Carpathians), with a confirmed presence point along its path. Although it is highly unlikely that it will actually cross using this corridor, human induced pressure may facilitate its dispersal. Most of these connections were performed in higher altitudes, crossing multiple river basins, thus yielding a low invasion risk potential.

In the north–north-western regions, the species was most likely connected through the Tisa River, since it was reported inhabiting riparian habitats in Hungary and Serbia [13,25]. The connectivity analysis showed an isolated section in the most northern part of Romania because the area is surrounded by mountain regions and its rivers are tributaries to the Tisa Basin. The area is connected through those rivers outside the processing limits of this study.

Nevertheless, we do not exclude in any of the analysed regions the possibility that the species was artificially introduced, and it did not occur through natural dispersion. With alien plant species, especially with the ornamental ones, we must add to the natural invasion risk, the human assistance spread over long distances through intentional planting in gardens [16] which may occur at any moment across different regions, increasing therefore its risk of invasion in new areas.

## 4. Materials and Methods

### 4.1. Study Area

The study area is in the southern and central parts of the Danube basin, where the species has expanded in recent years [24,26]. We selected a geopolitical boundary (Romania) due to the standardization of the occurrence collection sampling effort in the field within the above-mentioned research project, but also because part of the categorical geospatial datasets was locally available within a specific format. The landscape and climatic factors in the study area show strong heterogeneity and thus can be used as an optimal modeling boundary, although in some cases, the western and northern parts of the study area can only be connected through external major river systems, such as the Tisa River. The area comprises 5 biogeographical regions (Pannonian in the west, Continental in the central regions, Alpine in the mountain regions, Steppic in the eastern region, and Pontic in the south-eastern region), each showing a large variation of climatic, geological, and biological factors [75]. The western region is closer to the ocean and thus it receives more rainfall [76]. The southwestern region has a Mediterranean influence, with mild humid winters [74], while the eastern Steppic region has Arctic influences, with cold and dry winters [77]. Continental areas receive variate rainfall quantities, but in general, lower on the eastern and southern side of the Alpine regions [76]. The Carpathian Mountains (alpine regions) also act as a barrier towards air masses, registering significant rainfall [76]. The hydrographic network is almost completely tributary to the Danube River, with small exceptions in Steppic region [78]; thus, the dispersion of the species can theoretically be assured through this network. Regular flooding events are registered within these systems, especially during the spring, with higher intensities in smaller basins, closer to the alpine regions [79]. Climatic changes have been observed during the last decades, with increased droughts in the southern and eastern regions, but also with overall milder winters [80], increasing the need for invasive species distribution modeling approaches, which can predict their potential range shifts and help decision makers take appropriate action.

Between 2019–2021, we have identified new records of *H. scandens* in Romania (Figure 1), particularly on riverbeds, plains, and along roadsides. In the south-western region (Figure 1A), the species registered frequent occurrences, sometimes completely covering the riverbeds and alluvial plains of large rivers and their tributaries. Constant flooding events are recorded in the region because of hydro energy activities, which may favor the *H. scandens* dispersion. Within this region, *H. scandens* has also established in degraded areas which are dominated by reeds (*Phragmites australis*), or in abandoned industrial areas near watercourses. New data on *H. scandens* were also recorded from the southern region of the study area (Figure 1B), on two main tributaries of the Danube (Argeș and Dâmbovița), on highly modified riverbeds. The streams were used for irrigation and an extensive network of channels was developed and later abandoned, leaving behind optimal habitats for this alien species. *H. scandens* registered a higher frequency near watercourses, with only a few occurrences along roadsides and in degraded areas. Similar to the south-western region, the species has been observed occupying large sections of the riverbanks. Within this region, *H. scandens* registered its lowest altitudes observed within Romania, 11.5 m a.s.l.

The species is less abundant in the central and north-western regions of Romania (Figure 1C,D), with only a few recordings identified along roadsides and riverbanks. The highest altitude of our occurrences was located in Transylvania (Figure 1C), at 945 m a.s.l (Appendix A).

The dominating soil class occupied by the species in the mentioned regions is the anthrosol (Appendix A), a soil modified by human activity that caused profound changes in its properties [81]. The soil texture from the recently identified locations of *H. scandens* is mostly loamy-sand, loamy-clay, or variate (Appendix A). Although most observations of the study species were recorded in open spaces, there were some exceptions where *H. scandens* was identified in areas with a tree coverage varying from 64 to 82 % (Appendix A).

### 4.2. Environmental Variable Selection

To maximize the performance of the distribution modeling analysis and the accuracy of predictions, we relied on field observations and on literature data concerning the ecological requirements of *H. scandens* such as to select the most appropriate variables that may influence its distribution [82,83,84,85]. Thus, we considered a set of 17 environmental predictors related to the distribution of *H. scandens*, including climatic, topographic, pedologic, hydrologic, and human impact-related variables.

The selected climatic variables present basic constraints on plant distributions at global scale [12,86]: mean minimum temperature of the coldest month (Bio6), reflecting the exposure to winter cold; mean temperature of the warmest quarter (Bio10), which coincides with the growing season thermal regime; annual precipitation (Bio12), and the aridity and evapotranspiration indexes.

The bioclimatic variables were extracted from the WorldClim version 2.1 [87] at a spatial resolution of 30 arc seconds, while the aridity and evapotranspiration indexes were derived from Global Aridity Index and Potential Evapotranspiration (ET0) Climate Database v2–30 arc seconds [88].

We used a digital elevation model (DEM) with a spatial resolution of 25 m, from which we derived the slope and solar radiation, using ArcGIS v10.4. The DEM was obtained from EU-DEM v1.1 Copernicus (European Environment Agency—EEA) [89].

In order to evaluate the tree coverage from the forested areas, we used tree density index from the Copernicus EEA Platform developed for 2018, at a 100-m pixel resolution.

The Human Impact Index (HII) was obtained from the CIESIN (Center for International Earth Science Information Network)—Wildlife Conservation Society (Last of the Wild Data v2-2005 LTW2 Global Human Footprint Data set, at a resolution of 1 km^2^ [90].

We further obtained several vector data sets in order to create relevant spatial variables, which included roads and railroads, rivers, lakes, land use classes, soil class, soil texture, and soil structure. We downloaded these from the Open Street Map Contributors (OSM) at a scale of 1:1000; Corine Land Cover 2018 [91] at a scale of 1:100,000; and the European Soil Data Centre (ESDAC) (Atlas of Romania Soils Map 1:1,000,000) [92]. We further reclassified the CLC in order to obtain categories more appropriate for the species environmental preferences (Appendix A). The linear elements (roads, railroads, and rivers) were used to generate distance variables, which were corrected with the DEM in order to account for elevation variations, using ArcGIS v.10.4.

The final spatial resolution selected for the models was 100 m per pixel. This was achieved either by resampling and upscaling variables, such as the DEM derivates, or by downscaling using bilinear interpolation for the climatic datasets and nearest neighbor interpolation for the categorical datasets.

Before running the ESDM, we used the VIF function (car package) to test the correlation between the predictor variables, with R software v.4.1.2. The variables that registered values higher than 10 were eliminated from the model. The remaining 13 variables which were used for model training in the ESDM are listed in Table 2.

### 4.3. Species Occurrence Data

Occurrences of *H. scandens* were collected during the ongoing project “Adequate management of invasive species, POIM2014 + 120008” (49 recordings). In addition, we also added occurrences extracted from a literature review (10 recordings) [36,37,39,40,41,42,93,94].

### 4.4. Modeling the Species Suitable Habitats

In order to assess the *H. scandens* potential distribution across Romania, we used eight modeling algorithms within an ensemble model to account for the inter-model variability [95,96]. It is thought that the ensemble technique provides a better predictive performance compared to a single model approach [47]. We fitted the ensemble model using SSDM (‘SSDM’ R package) [96], with the following modeling algorithms: maximum entropy (MAXENT), classification tree analysis (CTA), multivariate adaptive regression splines (MARS), generalized linear model (GLM), generalized boosting models (GBM), generalized additive models (GAM), artificial neural networks (ANN), and random forests (RF). We performed the modeling using ten repetitions for each algorithm.

For the ensemble SDM, we coupled species occurrence data with 10,000 pseudo-absences which were randomly generated across the study area, with the SSDM package [96]. Models were built using a random subset of occurrence points (75% calibration dataset) and model performance was evaluated using the remaining occurrence points (25% validation dataset) [97,98]. The metric used to compute the binary map threshold was the TSS (true skill statistic) which maximizes the sum of sensitivity and specificity. Variable contribution to the model was calculated based on the Pearson correlation coefficient between the model with all variables and models where each variable was omitted in turn, using the SSDM package [96]. The best performing models were included in the ensemble SDM based on AUC > 0.75, further eliminating all the models which were below this threshold.

The following evaluation metrics were used to assess the performance of the models: receiver operating characteristic (ROC), area under the curve (AUC), omission rate, sensitivity, specificity, and true skill statistic (TSS). AUC, sensitivity, specificity, and TSS are metrics widely used to measure the model predictions accuracy [99,100,101]. The omission error rate constitutes a common measure of model performance, providing the proportion of positive test occurrence records falling outside the area predicted for the species, ranging from 0 to 1 [98].

### 4.5. Habitat Connectivity Analysis and Invasion Risk Assessment

To evaluate the potential invasive risk of the species, we have used binary results of the ensemble SDM to identify suitable habitat patches that can be colonized by its populations. This approach identifies a potential niche of the species, based on the observations and environmental variables which were used in the model training. We further removed patches smaller than 1 km^2^. We set this threshold based on our field analysis and literature review. We found that the species can consistently cover more than 500 m^2^ with a 100% coverage, and it can also extend for another 500 m^2^ with lower coverage percentages [20]. Using these patches as core areas (areas from which the species can colonize other habitats), we performed a habitat connectivity analysis to identify potential dispersion areas for the species and to evaluate its invasion risk. We used Linkage Mapper [102] as an extension to the ArcGIS (ESRI) environment. The inversed habitat probability outputs of the models were used as the resistance surface. This approach was used in various studies [59,103], and is considered advantageous in reducing expert opinion bias when choosing appropriate variables and especially while assigning the resistance threshold [59,104,105].

The linkage models resulted in least cost paths (LCP) that connect pairs of core areas through the resistance surface in areas with low resistance values, but also in a surface dataset which describes habitat permeability. We assessed the invasion risk based on the species ecology. According to the literature review, the seeds of *H. scandens* lack specific adaptations for dispersal, thus, its main means of natural dispersal is along the rivers and streams [106,107]. As mentioned within the EPPO Report, the magnitude of species spread in a river catchment is high, but to colonize other catchments, it needs human assistance [12]. Dispersion between basins is still possible, given the fact that the species also can use linear infrastructure elements. Therefore, we intersected the connectivity analysis results with the water catchments distribution, resulting in a sum of basins for each LCP corridor. We considered a high invasion risk for corridors within a single water catchment, medium risk for corridors which intersected two to five catchments, and low risk for all other values.

## 5. Conclusions

Our study showed a detailed approach on *H. scandens* potential habitat suitability and invasion risk through a connectivity analysis, identifying vulnerable areas which can be colonized by the species.

The results showed major rivers have been already colonized both in the upper and lower sections of their basins and potentially in-between, urging the need for immediate action plans regarding its containment.

This analysis represents a first step in managing this invasive species on the lower Danube Basin and can be used as a guideline for further field work research, but also for invasive management action plans.

## Figures and Tables

**Figure 1 plants-11-00857-f001:**
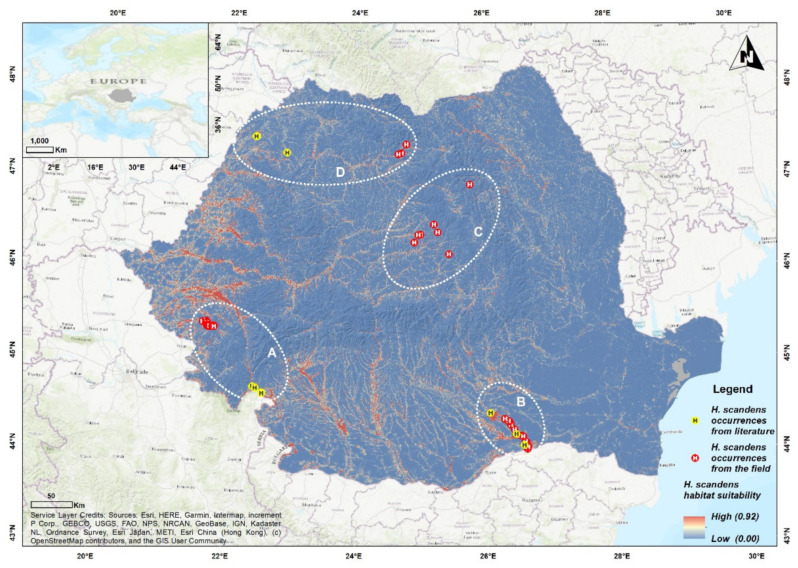
Updated distribution of *Humulus scandens* and habitat suitability modeling in (**A**) South-western, (**B**) Southern, (**C**) Central, (**D**) North-western Romania.

**Figure 2 plants-11-00857-f002:**
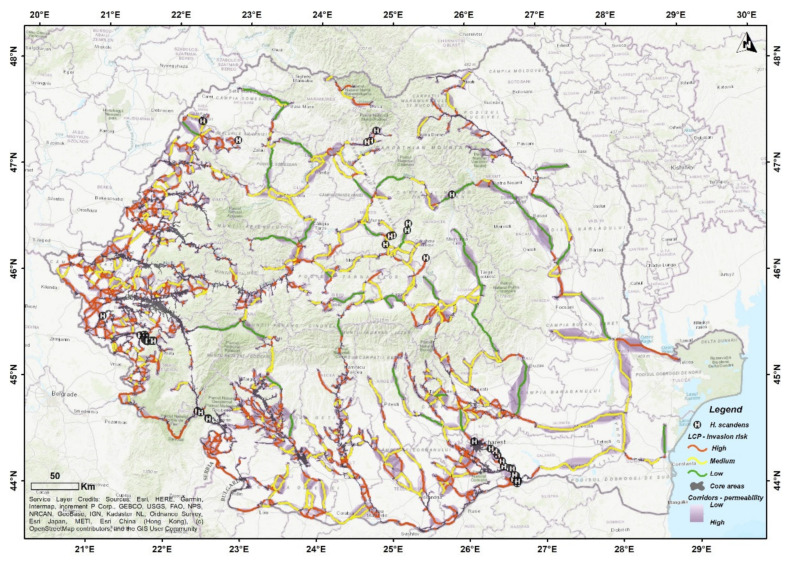
Habitat connectivity analysis and invasion risk of *H. scandens* in Romania.

**Table 1 plants-11-00857-t001:** Evaluation of each SDM and of the ESDM.

SDMs	SDM Performance				
	Sensitivity	Specificity	AUC	TSS	Omission Rate
MAXENT	0.8875	0.9020	0.8947	0.7895	0.1125
CTA	0.5313	0.9766	0.7539	0.5078	0.4688
MARS	0.8750	0.9111	0.8931	0.7861	0.1250
RF	0.9375	0.9309	0.9342	0.8684	0.0625
Ensemble SDM Performance (ESDM)					
ESDM	0.8078	0.9301	0.8690	0.7379	0.1921

**Table 2 plants-11-00857-t002:** Environmental variables used in the ensemble SDM and their contribution to the model.

Variables (Abbreviation)	Description	Units	Values Range (Min; Max)	Variables Contribution to the ESDM (%)
Bio6	Mean minimum temperature of the coldest month. One of the limiting factors of *H. scandens* in Europe has been reported to be low temperatures of the vegetation season [12].	°C	−89; −29	11.64
Bio12	Annual precipitation (relevant to vegetation growth). The stress caused by drought seem to be another limiting factor of *H. scandens* distribution across Europe. In dry environments, the species exhibits low invasion potential [16].	mm	525; 819	5.64
LUType	Land use types. The species can grow in disturbed areas including roadsides, old fields, and forest edges [12].	Categorical	-	13.79
DistLakes	Calculated path distance to lakes using DEM as a surface raster.	m	1; 11,826.96	3.98
DistWaterc	Calculated path distance to watercourses using DEM as a surface raster.	m	1; 1297.55	6.92
DistRoadsRailw	Calculated path distance to roads and railways using DEM as a surface raster. In its native range, *H. scandens* usually establish in disturbed habitats near settlements, roadsides, buildings, and waste deposits [13].	m	0; 1139.17	12.04
HII	Human Impact Index with values ranging from 1 to 100, estimating the relative anthropogenic impact. It is believed that *H. scandens* may have a preference for human-disturbed habitats [12].	#	14; 56	6.28
Slope	Derived from DEM	°	0.06; 24.33	6.88
SoilClass	Soil classes of Romania	Categorical	-	4.12
SoilText	Soil texture. In its native range it mostly establishes on loamy-sandy ground [13].	Categorical	-	17.42
SoilStruct	Soil structure	Categorical	-	2.55
SolarRad	Solar radiation derived from DEM	W/m^2^	1194.73; 1405.11	5.98
TreeDens	Tree density index, showing the percentage of tree cover. It is thought that *H. scandens* may prefer open-canopy areas or shaded areas for reasons unrelated to soil moisture content [22].	%	0; 82	2.72

## Data Availability

Not applicable.

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
