# Peer review of "Assessing the Invasion Risk of Humulus scandens Using Ensemble Species Distribution Modeling and Habitat Connectivity Analysis"

_plants, 2022, doi:10.3390/plants11070857_

Round 1
Reviewer 1 Report
Dear Authors,
The submitted manuscript titled „Assessing the invasion risk of Humulus scandens using ensemble species distribution modelling and habitat connectivity analysis”, reffering to important and still unsolved scientific problem of bilogical invasions, contains very valuable findings. However, I have found some mistakes, which in my opinion should be corrected before an eventual publication.
I have listed them below:
- In my opinion the aims of ivestigation (lines 83-86) should by placed at the end of chapter Introduction. Moreover, the description of a studied species might be enlarged with specific traits influencing on substantial invasivness (such as life span, seed production, seed dispersal).
- Lines 106-107 should be moved into Material and Methods
- I suggest to add the chapter containing information about study area, especially river network, topography, and others (i f it is possibile) which might influence on distribution and spread of Humulus scandens.
- Figures 1-2 are illegible.
- Discussion chapter should have a wider context. I suggest to compare the obtained results with publications reffering to climbers occuring in similar habitats which are invasive in Europe (e.g. Echinocystis lobata) or potentially invasive (Celastrus orbiculaus).
Please, look into below listed paper, which perhaps might be helpful in the manuscript improvement:
Kostrakiewicz-Gierałt et al. 2022. The Relationships of Habitat Conditions, Height Level, and Geographical Position with Fruit and Seed Traits in Populations of Invasive Vine Echinocystis lobata (Cucurbitaceae) in Central and Eastern Europe. Forests 13, 256. https://doi.org/10.3390/f13020256
Gudžinskas et al. 2020. Emerging invasion threat of the liana Celastrus orbiculatus (Celastraceae) in Europe. Neobiota 56:1-25.
Author Response
Dear Reviewer,
We would like to thank you for your time in revising our manuscript and for your suggestions for improving our paper. We are very glad that you think our manuscript contains valuable findings. Below, you will find our responses to your suggestions:
Point 1: In my opinion the aims of investigation (lines 83-86) should by placed at the end of chapter Introduction. Moreover, the description of a studied species might be enlarged with specific traits influencing on substantial invasiveness (such as life span, seed production, seed dispersal).
Response 1: We moved the aim of our investigation (lines 83-86) to the end of Introduction section (lines 113-116), as you suggested. This change required some other modifications to the remaining text, such as to be cursive.
We further extended the description of the study species from the Introduction, with other traits regarding its invasiveness (lines 66-71), as you also suggested. We would like to mention that features referring to the invasive character of the species are also mentioned within the other sections of the paper, where we thought it would be relevant (see lines 256-258, 274-276, 449-450, 462-464).
Point 2: Lines 106-107 should be moved into Material and Methods
Response 2: We moved the lines 106-107 to the Material and Methods section (lines 338-339), as you suggested. Furthermore, as the other Reviewer recommended, we moved the entire paragraph which describes the survey area from the Results (lines 107-134) to a new sub-section of Material and Methods that we created according to your suggestion, Study area (lines 339-363).
Point 3: I suggest to add the chapter containing information about study area, especially river network, topography, and others (if it is possible) which might influence on distribution and spread of Humulus scandens.
Response 3: We add a new chapter, Study area, where we briefly present Romania’s relevant features (including topography, climate, climate change effects, hydrography elements) in relation to the study species’ environmental preferences. As we already mentioned, this chapter also contains the description of the areas where we identified H. scandens in the field (lines 312-363).
Point 4: Figures 1-2 are illegible.
Response 4: We changed the fonts of the Legend and Grid of the Figures, as well as some colours we used, and we hope now they are eligible.
Point 5: Discussion chapter should have a wider context. I suggest to compare the obtained results with publications referring to climbers occurring in similar habitats which are invasive in Europe (e.g. Echinocystis lobata) or potentially invasive (Celastrus orbiculaus).
Response 5: We analysed the papers you suggested for extending the discussion section with a comparison between our results and other publications on invasive and potentially invasive climbing species from Europe that prefer similar habitats as Humulus scandens. We made a brief discussion using the information from these two articles that applied to our study (lines 221-234). Nevertheless, the results from these papers were not much comparable with ours, therefore we could not include a wider context on this subject.
We hope you agree with the changes we made and now you will find our paper as suitable for publication.
Reviewer 2 Report
I read this mauscript with interest. It is overall well written, even if English could be improved.
I have only one suggestion, which concerns moving the first portion of the results to the data section, since it describes the survey area and does not report actual results.
Author Response
Dear Reviewer,
We would like to thank you for your time in revising our manuscript and for your suggestion for improving our paper. We are very glad that you found our manuscript interesting and well-written overall. Below, we responded to your suggestion and explained the other changes we made in our paper.
Point 1: I have only one suggestion, which concerns moving the first portion of the results to the data section, since it describes the survey area and does not report actual results.
Response 1: We moved the first part of the results to the data section, as you suggested. Since this paragraph describes the survey area, and because the other reviewer suggested we should add a chapter with information regarding the study area, we thought it was appropriate to move the descriptive section from the results to this new chapter (lines 338-363). We hope you find this change as convenient and that it integrates fine within this chapter (lines 312-363).
In the revised version of our manuscript, you will also find other changes we performed according to the suggestions made by the other reviewer, such as enlarging the section regarding the study species from Introduction with other traits that influence its invasiveness (lines 66-71), and extending the context of the discussion chapter by comparing our results to other publications referring to climbers occurring in similar habitats as Humulus scandens, which are invasive or potentially invasive in Europe (lines 221-234).
We hope you agree with the changes we made and find our paper as suitable for publication.
Round 2
Reviewer 1 Report
Dear Authors,
In my opinion Your manuscript received sufficient improvements and I do not have any further remarks and suggestions.